# Investigation of Newly Synthesized Bis-Acyl-Thiourea Derivatives of 4-Nitrobenzene-1,2-Diamine for Their DNA Binding, Urease Inhibition, and Anti-Brain-Tumor Activities

**DOI:** 10.3390/molecules28062707

**Published:** 2023-03-16

**Authors:** Nasima Arshad, Uzma Parveen, Pervaiz Ali Channar, Aamer Saeed, Waseem Sharaf Saeed, Fouzia Perveen, Aneela Javed, Hammad Ismail, Muhammad Ismail Mir, Atteeque Ahmed, Basit Azad, Ishaq Khan

**Affiliations:** 1Department of Chemistry, Faculty of Sciences, Allama Iqbal Open University, Islamabad 44000, Pakistan; uzmarajpoot123@gmail.com (U.P.); overlord.scorpion6@gmail.com (M.I.M.); 2Department of Basic Sciences and Humanities, Dawood University of Engineering and Technology, Karachi 74800, Pakistan; pervaiz.ali@duet.edu.pk; 3Department of Chemistry, Quaid-i-Azam University, Islamabad 45320, Pakistan; asaeed@qau.edu.pk (A.S.); aahmed@chem.qau.edu.pk (A.A.); 4Restorative Dental Sciences Department, College of Dentistry, King Saud University, Riyadh 11545, Saudi Arabia; wsaeed@ksu.edu.sa; 5School of Interdisciplinary Engineering and Sciences (SINES), National University of Sciences and Technology (NUST), Islamabad 44000, Pakistan; fouzia@sines.nust.edu.pk (F.P.); basitazad50@gmail.com (B.A.); 6Healthcare Biotechnology Atta-ur-Rehman School of Applied Biosciences, National University of Sciences and Technology (NUST), Islamabad 44000, Pakistan; javedaneela19@asab.nust.edu.pk; 7Department of Biochemistry & Biotechnology, University of Gujrat, Gujrat 50700, Pakistan; hammad.ismail@uog.edu.pk; 8Texas A&M Health Science Center, Joe H. Reynolds Medical Build, College Station, TX 77843, USA; isaackhan1@tamu.edu

**Keywords:** bis-acyl-thiourea, DFT/docking analysis, spectral & electrochemical DNA binding, anti-brain tumor activity, urease inhibition

## Abstract

Bis-acyl-thiourea derivatives, namely *N,N’*-(((4-nitro-1,2-phenylene)bis(azanediyl)) bis(carbonothioyl))bis(2,4-dichlorobenzamide) (UP-1), *N,N’*-(((4-nitro-1,2-phenylene) bis(azanediyl))bis(carbonothioyl))diheptanamide (UP-2), and *N,N’*-(((4-nitro-1,2-phenylene)bis(azanediyl))bis(carbonothioyl))dibutannamide (UP-3), were synthesized in two steps. The structural characterization of the derivatives was carried out by FTIR, ^1^H-NMR, and ^13^C-NMR, and then their DNA binding, anti-urease, and anticancer activities were explored. Both theoretical and experimental results, as obtained by density functional theory, molecular docking, UV-visible spectroscopy, fluorescence (Flu-)spectroscopy, cyclic voltammetry (CV), and viscometry, pointed towards compounds’ interactions with DNA. However, the values of binding constant (*K*_b_), binding site size (n), and negative Gibbs free energy change (ΔG) (as evaluated by docking, UV-vis, Flu-, and CV) indicated that all the derivatives exhibited binding interactions with the DNA in the order UP-3 > UP-2 > UP-1. The experimental findings from spectral and electrochemical analysis complemented each other and supported the theoretical analysis. The lower diffusion coefficient (D_o_) values, as obtained from CV responses of each compound after DNA addition at various scan rates, further confirmed the formation of a bulky compound–DNA complex that caused slow diffusion. The mixed binding mode of interaction as seen in docking was further verified by changes in DNA viscosity with varying compound concentrations. All compounds showed strong anti-urease activity, whereas UP-1 was found to have comparatively better inhibitory efficiency, with an IC_50_ value of 1.55 ± 0.0288 µM. The dose-dependent cytotoxicity of the synthesized derivatives against glioblastoma MG-U87 cells (a human brain cancer cell line) followed by HEK-293 cells (a normal human embryonic kidney cell line) indicated that UP-1 and UP-3 have greater cytotoxicity against both cancerous and healthy cell lines at 400 µM. However, dose-dependent responses of UP-2 showed cytotoxicity against cancerous cells, while it showed no cytotoxicity on the healthy cell line at a low concentration range of 40–120 µM.

## 1. Introduction

Thiourea, an organosulfur compound, and urea have structural similarity with the exception that the –O atom is replaced by an –S atom. It is considered a very important reagent in organic synthesis. Thiourea and its derivatives are chemically versatile and biologically active compounds and possess a broad window of applications in medical chemistry such as antimicrobial agents, antibacterial agents [1,2,3], influenza virus inhibitors [4], anti-epileptic medications [5], anti-HIV medications [6,7], pesticides [8,9], and antioxidants [10,11]. Bioactive thioureas bear nitrogen as a hydrogen-binding area, sulfur as a complementary binding area, and substituents as auxiliary binding areas [12]. The sulfur- and nitrogen-containing core structure has an ability to develop hydrogen acceptor and hydrogen doner sites in the active pocket of various enzymes. Although thiourea and related sulfur- and nitrogen-containing compounds have limited selectivity for different targets and act like PAINS (pan-assay interference compounds), this is not true for all cases. Various thiourea-based pharmacophores selectively inhibit serval enzymes such as urease and glucosidase, and they act as antileishmanial agents. Thioureas also act as precursors for synthesis of various complexes and metal complexes [13]. The long-chain alkyl portion has an important role in adjusting lipophilic character in the molecule. Instead of acting as a lipophilic tuning portion, it has disadvantages due to its chain behavior. We can introduce a globular portion in the molecule for the purpose, such as adamantane [14]. The nitro group in the molecule has static charges and binds with DNA and other intracellular sites. The structures of some biologically active thiourea derivatives are provided in Figure 1 [15,16,17,18,19,20].

Bis-thioureas are compounds bearing two thiourea units. Such moieties are prone to notable pharmacological activities. Both symmetrical and unsymmetrical bis-thioureas have been employed in many therapeutics. For example, at nanomolar concentrations, phenyl-bis phenylthiourea (Figure 2a) exhibited cytotoxicity toward several cancerous cell lines [21], a polyamine analog of alkylated bis-thiourea (Figure 2b) exhibited antitumor activity by serving as a lysine-specific demethylase inhibitor [22], arylalkylpolyamino (bis)thiourea isosteres displayed antimalarial activity (Figure 2c) against *Plasmodium falciparum* [23], and a novel bis-thiourea derivative bearing an alkyl chain length (n = 10), (Figure 2d), showed marvelous anti-bacterial activity against *E. coli* ATCC 25922 [24]. These clinical and biological activities motivated us to synthesize nitro-phenylene derivatives of symmetrical bis-thioureas (UP-1–3) and to further study them for DNA binding, urease inhibition, and anticancer potentials.

## 2. Results and Discussion

### 2.1. Chemistry of the Synthesized Bis-Thioureas

Three bis-thiourea derivatives were synthesized according to the synthetic pathway depicted in Figure 1, which also displays the complete structures of the synthesized derivatives. Suitably substituted acid chlorides were converted to corresponding acyl isothiocyanate by addition of KSCN in acetone and then in situ followed by addition of an equimolar amount of 4-nitrobenzene-1,2-diamine to afford solid products which were recrystallized from ethanol to afford acyl thioureas (3a-c/UP1–UP3) in excellent yields (73–89%) and high purity.

The FTIR and NMR (^1^H-, ^13^C-) spectrum of each compound is provided in Appendix A. The FTIR spectral data revealed characteristic N-H stretching ranging from 3479 to 3187 and 3381 to 3030 cm^−1^, respectively, whereas carbonyl and thiocarbonyl stretching appeared in the range 1650–1695 cm^−1^ and 1240–1300 cm^−1^, respectively. In the case of UP-1, N-H stretching was found at 3479 cm^−1^ and 3381 cm^−1^, a C=O peak was found at 1697 cm^−1^, and a C=S peak was found at 1309 cm^−1^. In the ^1^H-NMR spectrum, two singlets integrating ^1^H each appeared in the range of 12.67–12.08 and 12.42–11.57, respectively. Alkyl protons appeared in an alkyl envelop in the range of 2.50–0.8 ppm. The ^13^C-NMR spectrum further affirmed the structural assignment by exhibiting the characteristic signals for the thioamide and amide carbons at 181.5 and 166.9 ppm, respectively, besides the aromatic or aliphatic, carbonyl, and thiocarbonyl carbons. Alkyl carbon signals appeared in the range of 36.12–14.40 ppm. 

### 2.2. In Silico Investigations

#### 2.2.1. DFT Studies

DFT/GGA:PBE was used to model and optimize structures of UP-1, UP-2, and UP-3 and to detect geometrical and electrical characteristics. The optimized structures of all the derivatives with symmetrical charge distribution and their mapping molecular electrostatic potential (MESP) surfaces mapped between −0.836 esu and 0.836 esu are illustrated in Figure 3. It is perceptible from Figure 3 that negative potenial is confined to –O, –Cl, and –S atoms, which intimates elecron transfer from these atoms. The –O, –Cl, and –S centers contribute as nucleophilic regions, whereas –C, –N and, –H atoms contribute as electrophilic regions, as indicated by red and blue color, respectively. Furthermore, the chemical transition was anticipated by frontier molecular orbital analysis (FMOs) [25,26]. The HOMO and LUMO orbitals are shown in Figure 3, and the their E_HOMO_ and E_LUMO_ ∆E values are provided in Table 1. The E_HOMO_ and E_LUMO_ values offer an idea of the nature of an electron-accepting or electron-donating compound, and, thus, a compound is deemed to be more electron accepting when the value of its E_LUMO_ decreases and more electron-donating when value of its E_HOMO_ increases. The computed values showed that the electron transfer in UP-3 is more viable as compared to UP-1 and UP-2 due to a smaller HOMO–LUMO gap; hence, UP-3 could be said to have highest reactivity amongst the investigated compounds.

The global indices/chemical descriptors such as ionization potential (I), electron affinity (A), softness (S), hardness (η), chemical potential (µ), and electronegativity (χ) were determined based on the HOMO and LUMO energy values by using the equations {(I = −E_HOMO_), (A = −E_LUMO_), (η = I − A/2), (μ = −(I + A)/2), (S = 1/2η), and (χ = (I + A)/2)}, and the results are presented in Table 1. The hardness was considered as the reactivity indicator. The greater hardness and lower softness of UP-1 as compared to UP-2 then UP-3 indicated that the deformation resistance of the electron cloud of the compound had increased, thus slightly decreasing reactivity. Most negative chemical potentials showed low interaction, and the data revealed that UP-3 had greater interaction, as it had greater value of chemical potential. Additionally, the softness of the compound UP-3 was found to be higher, which led to its higher interaction.

#### 2.2.2. Molecular Docking—DNA Binding and Anti-Urease Activity

The molecular mechanism between ligand–DNA and ligand–urease interactions could be simulated and interpreted by using a molecular docking approach. The compounds’ conformation (for UP-1–3) with minimum free energies and docked poses are given in Figure 4 and Figure 5, respectively, for their binding with DNA and urease. The binding free energy (∆G) and binding constant “*K*_b_” values evaluated for UP-1–3–DNA and for UP-1–3–urease interactions are provided in Table 1. 

The molecular docking analysis revealed interaction of all the compounds with DNA by a mixed mode including partial intercalation and groove binding. The LigPlots indicated 2D interactions of all compounds with DNA, as shown in Figure 4. For UP-1 and UP-2, the ligands showed two hydrogen (H-) bonding interactions. The –S atom of UP-1 developed one H-bonding interaction with the DNA base pair (DCA3), and its –O atom showed an H-bonding interaction with the DNA base pair (DGA4). Moreover, DNA base pairs (DGB22) exhibited two H-bonding interactions with the –O atom and –S atom of UP-2. The ligand UP-3 was found to be more potent, as five interactions were observed in three different ways. The H-bonding interactions of the –S atom of the ligand was observed with DNA base pair (DGB16), and the NH group of the ligand showed two H-bonding interactions with DNA base pair (DGB16). The carbonyl group of the ligand showed two H-bonding interactions with DNA base pair (DCA11). The *K*_b_ and ∆G values are provided in Table 1, and the values revealed comparatively stronger and spontaneous binding of UP-3 with the DNA. The molecular docking results further verified greater reactivity of UP-3, as is obvious from DFT, which gives it a higher binding constant than UP-1 and UP-2.

The urease enzyme also displayed 2D interactions with all the three compounds, as shown in Figure 5 (LigPlots). For UP-1, the –S group showed two H-bonding interactions: one with residue PheC1565 and the other with residue LysC1443 of the urease. In the case of UP-2, only one H-bonding interaction was detected between the “N-H” group of the ligand (UP-2) and enzyme residue ValC1471. For UP-3, the –S atom of the ligand showed one H-bonding interaction with the TyrC1473 residue of enzyme. The binding data, displayed in Table 1, pointed to stronger binding of UP-1 compared to the other compounds with urease enzyme with comparatively higher *K*_b_ and more negative ∆G values. 

### 2.3. DNA Binding—Experimental Investigations

#### 2.3.1. Spectral UV-Visible and Fluorescence Studies

Before performing the titration experiments, absorption and emission spectra of each compound (UP-1-3) were determined for a 2 × 10^−5^ M solution using a UV-visible and fluorescence spectrophotometer, respectively. The ε (molar extinction coefficient) value of each compound was determined by measuring the absorbance spectrum at increasing concentrations and then plotting absorbance vs. concentration to get the slope that represented the ε value at 1 cm path length in Beer’s equation (A = εcl) (see Appendix A). The obtained values of 75,100, 127,000, and 11,900 M^−1^ cm^−1^ for UP-1, UP-2, and UP-3, respectively, revealed that the operative transitions within the compounds were π-π^*^ transitions. 

Absorption spectral responses of each compound after DNA titrations are given in Figure 6a–c. In the presence of DNA, the spectral responses of UP-1 and UP-3 showed a hypochromic effect on the compound’s peak in 350–420 nm and 300–430 nm ranges, respectively, while for UP-2, a hyperchromic effect was noted in the 260 –300 nm range. The observed hypo- and hyperchromicity was generally associated with the structural changes in the DNA that arise due to the conformational variation and destruction in the double helix, respectively, in the presence [27]. The literature has also reported that such effects along with wavelength shifts (blue/or red) are indicative of compound–DNA interactions that could most probably occur through intercalation [28,29,30]. The presence of an isosbestic point indicated the establishment of an equilibrium between the intercalated complex and free compound and that no other species are present [29].

The emission spectra of UP-1, UP-2, and UP-3 were recorded in the range of 388–396 nm at the excitation wavelength of 368 nm, 315–400 nm at the excitation wavelength of 290 nm, and 300–380 nm at the excitation wavelength of 280 nm, respectively (see Appendix A). The spectral responses of each compound after DNA titrations are given in Figure 6d–f, which show enhancement in the peak intensity of UP-1 and quenching in the peak intensities for UP-2 and UP-3 upon DNA additions. Such spectral variations have been reported for the binding interactions and formation of compound–DNA adducts that were presumably linked via the intercalative binding mode [31,32].

The extent of binding interactions was further evaluated by using absorbance and emission intensity data of spectral responses of the derivative itself and in the presence of DNA in the following equations [31,32].
(1)$$\frac{A_{o}}{A-A_{o}}=\frac{\varepsilon_{G}}{\varepsilon_{H-G}-\varepsilon_{G}}+\frac{\varepsilon_{G}}{\varepsilon_{H-G}-\varepsilon_{G}}\frac{1}{K_{b}\left[DNA\right]}$$
(2)$$\log\frac{F-F_{o}}{F}=\log K_{b}+n\log\left[DNA\right]$$

Plots of A_o_/A − A_o_ vs. 1/[DNA] and log [F − F_o_/F_o_] vs. log [DNA] are shown in Figure 7, where intercept to slope ratio (Equation (1)) and antilog of intercept (Equation (2)), respectively, were used to find the values of binding constant “*K*_b_”. Binding site size (n) of the complex (compound–DNA) was estimated as the slope value in Equation (2). The Gibbs free energy change (∆G) value was obtained by using the “*K*_b_” value in the Equation. ∆G = −RTln *K*_b_ (Van ’t Hoff equation). The evaluated binding parameters are given in Table 2. The ∆G values and binding order range (*K*_b_; 10^3^–10^5^) reflected spontaneous and substantial binding of all the bis-thiourea derivatives with DNA [33,34]. The greater binding site sizes (n > 1) exhibited intercalation along with the availability of additional sites for developing other reversible interactions, i.e., electrostatic or groove binding (see Table 2) [31,32,33].

#### 2.3.2. Electrochemical CV Studies

A 0.1 V/s scan rate was used to record the cyclic voltammograms (CV) for the three bis-thiourea derivatives individually and for each compound–DNA adduct. Individual scanning of the compounds showed a reduction peak with quasi reversibility for UP-1 (∆E_p_ = 0.33 V) and irreversibility for UP-2 (E_pc_ = −0.815 V) and UP-3 (E_pc_ = −0.750). However, during compound–DNA adduct formation while increasing DNA concentrations, all the derivatives showed an irreversible nature. The shifting of quasi reversibility to irreversibility could be related to the oxidizing groups (i.e., oxygen of C=O and sulfur of C=S) of UP-1 that interacted with DNA, as also verified by molecular docking. Therefore, its oxidation peak disappeared after the addition of DNA. The reduction peak current of UP-1, UP-2, and UP-3 significantly dropped down to 50%, 53%, and 70%, respectively, along with a positive shift (less negative potential) in E_½_ for UP-1 and in E_pc_ for UP-2 and UP-3 (see Figure 8a–c). These observations revealed the fact that all the synthesized bis-thiourea derivatives interacted with the DNA, preferably via intercalation [35]. The compound–DNA adduct was also assured by calculating the diffusion coefficient (D_o_) of the derivatives alone and after DNA addition using linear dependency of peak current with scan rate in the Randles–Sevcik equation {I_𝑝_ = 2.99 × 10^5^ n(αn_α_)^1/2^𝐴_0_𝐶_0_^*^𝐷_0_^1/2^υ^½^} (see Appendix A). The D_o_ values (cm^2^ s^−1^) of UP-1, UP-2, and UP-3 decreased from 6.55 × 10^−9^, 1.79 × 10^−8^, and 4.48 × 10^−9^ to 2.32 × 10^−9^, 3.05 × 10^−9^, and 1.12 × 10^−9^, respectively, after DNA addition, which further confirmed the formation of massive compound–DNA complexes.

Using Equations (3) and (4) [36], *K*_b_ and n were evaluated, and the plots are provided as d–f and g–i, respectively, in Figure 8.
(3)$$I_{p}^{2}=\frac{1}{K_{b}\left[DNA\right]}\left(I_{P_{o}}^{2}-I_{P}^{2}\right)+I_{P_{o}}^{2}-\left[DNA\right]$$
(4)$$\frac{I-I_{DNA}}{I_{DNA}}=\frac{K_{b}\left[DNA\right]}{2n}$$

The binding parameters’ values are provided in Table 2, and the trends in the values complemented the results obtained from spectral findings.

#### 2.3.3. Viscosity Studies

The binding modes of interaction were additionally verified by monitoring the changes in DNA viscosity in the presence of increasing compound concentration. Graphs were plotted between relative specific viscosities against compound–DNA concentration ratios in Figure 9. Generally, the viscosity of the DNA solution is enhanced by increasing the concentrations of the compound [31]. This could be attributed to DNA base pairs stretching due to the accommodation of the intercalating structure of the compound, and the size enlargement resulted in the enhancement of DNA viscosity [31,37,38]. The viscosity measurements in the present work revealed that initially, DNA viscosity increased as the compound concentration increased, but after certain additions, no further rise in DNA viscosity was observed. Such changes in the DNA viscosity confirmed that all the derivatives adopted a mixed binding mode for their interactions with DNA [31,37], which most likely could be partial intercalation and groove binding as pridicted during docking studies. 

### 2.4. Anti-Urease Activity Studies

Newly synthesized derivatives were tested for their ability to inhibit urease enzyme at three different concentrations, and results of percentage inhibition are presented in Figure 10. The IC_50_ values were evaluated as mean (n = 3) ± standard deviation (SD). The percentage inhibition of all the compounds at 100 µM was evaluated to be 77.83%, 72.81%, and 69.83% for UP-1, UP-2, and UP-3, respectively. The reported literature validated that a compound with an IC_50_ < 1 µM has potent activity; compounds with values in the ranges 1–20 µM, 20–100 µM, and 100–200 µM have strong, moderate, and low activity, respectively; and compounds with an IC_50_ >200 µM have no activity [39]. The results of IC_50_ indicated strong activity of all the compounds against the urease enzyme. However, the highest activity was shown by compound UP-1, with an IC_50_ value of 1.55 ± 0.0288 µM, followed by compounds UP-2 and UP-3, with the values 1.66 ± 0.0179 and 1.69 ± 0.0162 µM, respectively. Moreover, thiourea, which was used as a positive control, revealed prominent enzyme inhibition activity, with a percentage inhibition of 90.91% at 100 µM and an IC_50_ value of 0.97 ± 0.0371 µM. Additionally, all the compounds showed concentration-dependent activity. The graphs for the calculation of IC_50_ values are provided as Appendix A. 

### 2.5. Cytotoxicity Studies against Brain Tumor and Normal Cell Lines

Cytotoxicity activities of the derivatives were evaluated by using MG-U87 and HEK-293 cell lines, and concentration-dependent comparison graphs are provided in Figure 11. The results obtained from MTT analysis on the MG-U87 cell line indicated the average percent cytotoxicities for all concentrations (40, 80, 120, 200, and 400 µM) of compound UP-1, which were 27.8%, 36.5%, 40.1%, 43.7%, and 52.9%, and the cytotoxicity values for HEK-293 cells were 21.8%, 44.5%, 52.9%, 59.2%, and 69.9%, respectively. Cytotoxicity results indicated that HEK-293 cells did not show any tolerance to the increasing concentration of UP-1, and UP-1 was more cytotoxic to the normal cells. Greater cytotoxicity to healthy cells showed that UP-1 could not have significant anticancer potential against MG-U87 even at its highest concentration of 400 µM, as cytotoxic activity of this compound was much greater against healthy cells at the same concentration. Overall, the trend of percent cytotoxicity increased with increases in compound’s concentration for both the cancereous and healthy cell line, where, for latter, this compound showed greater cytotocity.

No cytotoxic effect was found on HEK-293 in the presence of first three concentrations of UP-2, while the last two concentrations showed 24.1% and 27.1% cytotoxicity, respectively. However, UP-2 showed concentration-dependant cytotoxicity for MG-U87, and the values were found to be 10.9%, 12.2%, 29.4%, 39.2%, and 43.09%, respectively, for the concentrations ranging from 40 to 400 µM. The percent cytotoxicty values of UP-3 were found to be 17.31%, 37.4%, 44.3%, 60.7%, and 60.9% and 17.18%, 18.56%, 37.0%, 41.14%, and 56.9% for HEK-293 and MG-U87 cell lines, respectively. 

These results indicated that compounds UP-1 and UP-3, at all concentrations, are toxic to both healthy (HEK-293) and cancereous (MG-U87) cell lines. However, no cytotoxicity of UP-2 at low concentrations (40–120 µM) and comparitively low cytotoxicity at 200 to 400 µM showed a promising impact of UP-2 on healthy cells. However, percent cytotoxicity of UP-2 for MG-U87 was found to be comparitively less (43.09%) than that evaluated for UP-1 (52.9%) and UP-3 (56.9%) at highest selected concentration of 400 µM. Using GraphPad PrismV8, the IC_50_ values for UP-1, UP-2, and UP-3 for MG-U87 and HEK-293 cell lines were found to be 2.496 ± 0.0324 µM, 2.664 ± 0.1298 µM, and 2.459 ± 0.0656 µM as well as 2.096 ± 0.0487 µM, 2.856 ± 0.2027 µM, and 2.220 ± 0.0707 µM, respectively. The graphs for the calculation of IC_50_ values are provided as Appendix A. 

The IC_50_ values were found to be comparatively less from our previously reported work on isatin derivatives, where IC_50_ values for IST-2 and IST-4 derivatives were reported to be 3.07 ± 9.47 and 14.60 ± 2.49, respectively, for the HuH cancer cell line [31]. The IC_50_ values in the current studies were also compared with the IC_50_ values of thiourea derivatives for different cancerous cell lines [40,41,42], which are provided in Table 3. This comparison further indicates that the synthesized bis-thiourea derivatives in the current studies have lower or comparable IC_50_ values to other reported thiourea derivatives. However, dose-dependent cytotoxicity indicated no cytotoxicity (0%) at lower concentrations while about 37% cytotoxicity at the highest concentration of 400 µM of the compound UP-2 towards the healthy cell line. 

Since cancer therapeutics is a challenging filed, the latest literature focuses on the use of nano-carrier-conjugated drugs for the targeted delivery of the drugs to the cancer cells such as gold nano cages, carbon rods, chitosan nanoparticles, lipid nano carriers, as well as natural products [43,44,45]. The compounds tested in vitro in the current study can be further conjugated in nanoparticles for more efficient cancer treatments. 

## 3. Experimental

### 3.1. Materials and Methods

Highly pure chemicals and reagents were used during synthesis and other experimental steps. Standard approaches were used to purify and dry the solvents. The NMR spectrum of each compound and an internal reference TMS (tetra-methyl silane) was determined by using a deuterated solvent (DMSO). Chicken blood was used to extract dsDNA through the Falcon protocol, which was dissolved in deionized water to carry out compound–DNA binding experiments. The stock DNA solution was further diluted, and its absorbance at λ_max_ of 260 nm was measured. The DNA concentration 4.79 × 10^−5^ M was obtained by using the molar extension coefficient (ε) value of 6600 cm^−1^ M^−1^ and a path length (Ɩ) of 1 cm in Beer’s equation. The absorbance ratio at 260 nm to 280 nm was found to be 1.87, which assured DNA purity [31,35]. For DNA binding studies, the quartz cells and a double-walled cell containing working {glassy carbon (GC), area; 0.070 cm^2^}, reference {silver/silver chloride (Ag/AgCl), filling: 3.0 M KCl}, and counter (99.99% Pt wire, diameter; 0.5 mm) electrodes were used in spectral and cyclic voltametric (CV) experiments, respectively. These cells were kept at 37 °C for a few minutes using a temperature controller. Prior to each CV experiment, the cleaning of the GC electrode surface with an alumina slurry followed by ultrasonication for 30–50 s and flushing out of oxygen from the cell by Ar gas (99.99%) purging for at least 6–8 min were conducted as compulsory steps. 

### 3.2. Instrumentations

The synthesized materials were characterized for their structures by melting point determination by Gallen Kamp M.P. apparatus and by Excalibur FT-IR spectrometer (Bio-Rad, Bruker, Billerica, MA, USA, FTS 300 MX), ^1^H-NMR (Bruker 300 MHz NMR), and ^13^C-NMR (75 MHz NMR channel). DNA binding studies were performed on UV-Vis (Shimadzu1800-, TCC-240, Tokyo, Japan), fluorescence (F-7000 model FL2133-007) spectrophotometers, electrochemical workstation (AUTOLAB PGSTAT-302, GPES version 4.9, Metrohm, The Netherlands), and digital viscometer (Schott Gerate automated, Mainz, Germany, AVS 310). 

### 3.3. Synthesis of Nitrophenylene Derivatives of Symmetrical Bis-Acyl-Thiourea

Potassium thiocyanide (5.5 mmol) solution was prepared in dry acetone (15 mL), and substituted acid chloride (5 mmol) (1) was added dropwise under an inert environment for 1.5 h at 70 °C. Milky color of solution indicated the formation of an acyl thiocyanate intermediate (2). After cooling at room temperature, the solution of 4-nitrobenzene-1,2-diamine (2.5 mmol) in acetone was added dropwise into it using an additional funnel under an inert environment in 15-mints. A continuous stirring of the reaction mixture was carried out for 9 h, and progress of the reaction was monitored on a TLC plate. As the reaction completed, the mixture was transferred into a beaker with crushed ice. The desire product (3) was precipitated as a yellow solid, which was then filtered, washed with cold water, dried, and recrystallized from ethanol.

### 3.4. Characterization Data


*N,N’-(((4-nitro-1,2-phenylene)bis(azanediyl))bis(carbonothioyl))bis(2,4-dichlorobenzamide)—(UP-1)*


Yellow solid, m.p = 257–259 °C, yield = 73%, R_f_ = 0.52 (chloroform:methanol 4:1); FT-IR (ν cm^−1^): 3479 (NH), 3381 (NH), 3159,3062 (Ar-CH), 1697,1624 (C=O), 1583 (C=C), 1523 (thioamide I), 1492 (thioamide II)^1^ H NMR (DMSO-d_6_, 300 MHz,); δ (ppm): 12.08 (s, 1H, NH), 11.57 (s, 1H, NH), 8.09 (s, 1H, Ar-H), 7.79 (d, 1H, J = 9.3Hz, Ar-H), 7.74 (s, 1H, Ar-H), 7.64 (d, 1H, J = 8.1 Hz, Ar-H), 7.59 (d, 1H, J = 8.1 Hz, Ar-H), 6.81 (d, 1H, J = 9.3 Hz, Ar-H); ^13^C NMR (75 MHz DMSO-d_6_) δ (ppm) 181.57 (C=S), 166.93 (C=O), 151.59, 136.37, 135.70, 133.72, 131.91, 131.33, 129.73, 127.79, 126.07, 125.35, 121.95, 114.50 (Ar-C) Anal. Calcd. for C_22_H_13_C_l4_N_5_O_4_S_2_: C, 42.80; H, 2.12; N, 11.34; S, 10.39. Found: C, 42.81; H, 2.14; N, 11.32; S, 10.37. HRMS Caled for C_22_H_13_C_l4_N_5_O_4_S_2_+H: 616.9134. Found 616.9130.


*N,N’-(((4-nitro-1,2-phenylene)bis(azanediyl))bis(carbonothioyl))diheptanamide—UP-2*


Light yellow crystalline solid, m.p = 175–176 °C, yield = 84%, R_f_ = 0.68 (*n*-hexane:ethyl acetate 4:1), FT-IR (ν cm^−1^): 3197 (NH), 3030 (Ar-CH), 2925,2860 (CH_2,_ CH), 1698,1621 (C=O), 1580 (C=C), 1525 (thioamide) ^1^H NMR (DMSO-d_6_, 300 MHz,); δ (ppm): 12.67 (s, 1H, NH), 12.41 (s, 1H, NH), 11.68 (s, 1H, NH), 11.67 (s, 1H, NH), 8.79 (d, 1H, J = 2.7 Hz, Ar-H), 8.45 (d, 1H, J = 9 Hz, Ar-H), 8.21 (q, 1H, J = 2.7 Hz and J = 9.0 Hz, Ar-H), 2.51–2.41 (m, 4H), 1.58 (t, 2H, J = 6.9 Hz CH_2_), 1.27 (m, 12H), 0.87 (t, 6H, J = 6.3 Hz, CH_3_); ^13^C NMR (75 MHz DMSO-d_6_) δ (ppm) 181.11 (C=S), 175.8 (C=O), 175.64 (C=O), 144.84, 139.96, 133.81, 126.14, 122.27, 122.10 (Ar-C), 36.22, 31.51, 28.60, 24.80, 24.75, 22.44, 14.40 (Alkyl chain C) Anal. Calcd.for C_22_H_33_N_5_O_4_S_2_: C, 53.31; H, 6.71; N, 14.13; S, 12.94. Found: C, 53.34; H, 6.75; N, 14.11; S, 12.92. HRMS Caled for C_22_H_33_N_5_O_4_S_2_+H: 495.1974. Found 495.1971.


*N,N’-(((4-nitro-1,2-phenylene)bis(azanediyl))bis(carbonothioyl))dibutyramide—UP-3*


Light yellow crystalline solid, m.p = 160–162 °C, yield = 89%, R_f_ = 0.57 (*n*-hexane:ethyl acetate 4:1), FT-IR (ν cm^−1^): 3197 (NH), 3032 (Ar-CH), 2962,2906 (CH_2,_ CH), 1698,1623 (C=O), 1598 (C=C), 1520, 1477 (thioamide) ^1^H NMR (DMSO-d_6_, 300 MHz,); δ (ppm): 12.67 (s, 1H, NH), 12.42 (s, 1H, NH), 11.68 (s, 2H, NH), 8.78 (s, 1H, Ar-H), 8.44 (d, 1H, J = 8.7 Hz, Ar-H), 8.20 (d, 1H, J = 8.4 Hz, Ar-H),2.50–2.40 (m, 4H, CH_2_), 1.58 (m, 5H, CH_2_), 0.91–0.90 (d, 7H, 2(CH_3_)); ^13^C NMR (75 MHz DMSO-d_6_) δ (ppm) 181.12 (C=S), 175.7 (C=O), 175.5 (C=O), 144.89, 139.97, 133.85, 126.25, 122.30, 122.16 (Ar-C), 36.2, 29.3, 14.4 (Alkyl chain C) Anal. Calcd.for C_16_H_21_N_5_O_4_S_2_: C, 46.70; H, 5.14; N, 17.02; S, 15.58. Found: C, 46.72; H, 5.12; N, 17.04; S, 15.56 HRMS Caled for C_16_H_21_N_5_O_4_S_2_ +H: 411.1035. Found 411.1032.

### 3.5. DFT and Docking—Theoretical Procedures

Quantum chemical studies were carried out using Amsterdam Density Function Modeling Suite, and ADF builder was used to generate structures and to visualize graphics [46,47]. All compounds were optimized at GGA: PBE method with DZ basis set. The GGA: PBE is a more popular and reliable theoretical method in recent years to elucidate the properties of a compound’s structure due to its accuracy and economical cost [48]. The computation of FMO (frontier molecular orbitals), bandgap, and MEP (molecular electrostatic potential) were also made at GGA: PBE/DZ level of theory. FMOs were computed through electronic energy levels, and MEP was computed by XC potential iso surfaces. Both were visualized using SCM ADF viewer.

The structural optimization of compounds (UP-1–3) was conducted using version 2015.10 of MOE (Molecular Operating Environment) at MOPAC 7.0 level of theory. Following the geometry relaxation, the structures of all the compounds were constructed and collected into the MOE database. The DNA and urease enzyme crystallographic structures having PDB ID: 1BNA and PDB ID: 1EJU and having resolution of 1.9Å and 2.0Å, respectively, were fetched from the Protein Data Bank for molecular docking simulations with UP-1, UP-2, and UP-3 [49,50]. The molecules of H_2_O that were attached with the DNA base pairs (1BNA), heteroatoms, and co-crystallized ligands were removed. Similarly, H_2_O molecules attached to urease enzyme (1EJU) base pairs were also removed using sequence editor of MOE. Additionally, 1BNA and 1EJU were protonated and optimized expending protonate-3D menu. Coordinates of 1BNA and 1EJU were relaxed using AMBER forcefield and semi-empirical PM3 approaches for docking analysis purpose. For the optimal computation, the energy and stability of the relaxed coordinates were kept minimal, and the best scoring functions were computed. For docking, all optimized structures were submitted to systematic molecular docking utilizing 1BNA and 1EJU as default parameters (RMS gradient = 0.01 kcal mol^−1^) and using Site Finder to locate 1BNA and 1EJU active sites. Several docking runs might achieve the final docking positions as perfectly as feasible. The energy of the interaction of compounds with 1BNA and 1EJU were evaluated at each stage of the simulation. Rest settings were maintained as default [25].

### 3.6. DNA Binding—Experimental Procedures

Individual absorption and emission spectra and the cyclic voltammogram of each compound (UP–1-3) were recorded for their optimized concentration (2 × 10^−5^ M). Then DNA titrations, upon the compound’s fixed concentration, were carried out in spectrophotometric (UV-visible and fluorescence) and cyclic voltametric (CV) experiments using DNA concentrations ranging from 10 to 70 μM at pH (7.0) and at physiological temperature of 37 °C [51]. The cells used in these experiments were kept at rest for a few minutes before each run to assure an equilibrium for the compound–DNA complex. UV-visible experiments were run within 200–500 nm, while 200 nm and 900 nm, respectively, were the EM start and EM end wavelengths in fluorescence experiments. The cyclic voltametric experiments were run within −2 to +1 V at a scan rate of 0.1 V/s, and for the determination of D_o_ (diffusion coefficient), the scan range of 0.03– 0.13 V/s was used with a difference of 0.02 V/s before the next scan. In viscosity experiment, DNA viscosity (η_o_) at an optimized concentration of 10 µM was measured, and then the small variations in the DNA viscosity (η) after increases in the compound’s concentration (10–70 µM) were monitored. The observed fractional changes in DNA viscosity were used to determine the binding modes of interaction. 

### 3.7. Anti-Urease Assay

The anti-urease activity of the compounds was measured by determining the amount of free ammonia produced as described earlier [52]. The experiment was performed by mixing 10 µL of enzyme (0.1 U/per reaction), 30 μL of each concentration (100, 75, 50, 25, and 12.5 µM) of the compound, and 50 μL of buffer at pH 8.2 consisting of 100 mM urea, 0.01 M LiCl_2_, 1 mM EDTA, and 0.01 M K_2_HPO_4_. Reaction mixtures were incubated at 37 °C for 15 min in a 96-well plate. Then 50 μL of phenol reagent (0.005% sodium nitroprusside and 1% phenol) and 50 μL of alkali reagent (0.1% NaOCl + 0.5% NaOH) were added to each well, and plates were incubated at 37 °C for 50 min. The assay was performed in triplicate, and absorbance (A) was recoded at 625 nm using a microplate reader. The anti-urease activity of each compound was calculated in percentage inhibition using the following formula, and IC_50_ was calculated using GraphPad PrismV8.
% inh. = (A_control_ − A_sample_/A_control_) × 100

### 3.8. Cell Line Assay

The MTT analysis is the most common type of assay involving cell lines. MTT (3-(4,5-dimethyl-2-thiazolyl)-2,5-diphenyl-2H-tetrazolium bromide) is a dye used for the measurement of in vitro cell proliferation. Tetrazolium salts have been widely used tools in cell biology for determining the metabolic activity of cells ranging from microbial origin to mammalian cells [53]. For this purpose, the assay was performed on two cell lines: MG-U87 (malignant glioma cell line) and HEK-293 (human embryonic kidney cell line). Cells were maintained in Dulbecco’s Modified Eagle Medium (DMEM) (Gibco, Life Technologies, Waltham, MA, USA, catalogue 31800-022) and supplemented with 10 % fetal bovine serum (Gibco, Life Technologies, catalogue 16050) and 1% penicillin–streptomycin (Gibco, Life Technologies, catalogue 00580). The exponential growth of cells was counted. Then, in triplicate, 10,000 cells/well were plated in *Nunc* MicroWell 96-well microplates (Fisher Scientific, Roskilde, Denmark) by keeping the cells’ volume at 100 µL/well. The plates were incubated at 37 °C for 24 h in a 5% CO_2_ incubator. 

The compounds (UP-1, UP-2, and UP-3) were dissolved in 1 mL of 10% DMSO solution to obtain five concentrations 40, 80, 120, 200, and 400 µM (10 µL/mL, 20 µL/mL, 30 µL/mL, 50 µL/mL, and 100 µL/mL). These concentrations were then added, separately, into 96-well plates to get ~200 µL/well as the final volume. Moreover, all concentrations were individually tested on both HEK-293 and MG-U87 cell lines in triplicate. Control wells contained solvent control (without drug) and blank media (without cells). The plates were kept for 48 h in a 5% CO_2_ incubator at 37 °C. Subsequently, 5.0 mg/mL MTT per 1.0 mL of PBS was prepared, and from this solution, 15 µL was added to each well and, at 37 °C, it was incubated for 3 h to microscopically visualize the formazan crystals. The solution from the wells was discarded after the formation of formazan crystals. Then the plates were kept for a short period at room temperature, and the crystals were dissolved in 100 µL DMSO in each well. Lastly, the absorbance measurements of the cells were done at 550 nm.

## 4. Conclusions

Three bis-acyl-thioureas UP-1, UP-2, and UP-3 were synthesized and characterized by different spectral techniques (FT-IR, ^1^H-NMR, and ^13^C-NMR) that confirmed the compounds’ structures. These derivatives were further investigated for DNA binding, anti-urease, and anti-brain-tumor activities. Theoretical and experimental studies indicated that all the derivatives interacted significantly and spontaneously with DNA via partial intercalation and groove binding. However, the binding parameters (*K*_b_, ∆G, and n) were evaluated in the order UP-3 > UP-1 > UP-1. The formation of bulky compound–DNA complex was further confirmed by CV studies where the determined values of the diffusion coefficient (D_o_) were evaluated to be smaller as compared to compounds’ D_o_ values without DNA. The binding parameters (*K*_b_*;* 6.73 × 10^5^ M^−1,^ ∆G; −33.25 kJ mol^−1^) for UP-1 obtained from docking studies were found to be comparatively greater than other compounds for their interaction with urease enzyme. All the compounds showed strong anti-urease activity in the order UP-1 > UP-2 > UP-3, which matched with the molecular docking results. Cytotoxicity activity of all the compounds was tested against brain tumor (MG-U87) and normal (HEK-293) cell lines. Compounds UP-1 and UP-3 showed greater cytotoxicity on both healthy and cancerous cells, while cytotoxicity of UP-2, in comparison to brain tumor cells, was found less (at concentrations >200 µM) to none (at concentrations <200 µM) on normal cells, thus showing a comparatively promising effect. These studies may help to enlighten the role of new bis-thiourea derivatives for the exploration of their drug candidacy.

## Data Availability

Not applicable.

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
