# Peer review of "Investigation of Newly Synthesized Bis-Acyl-Thiourea Derivatives of 4-Nitrobenzene-1,2-Diamine for Their DNA Binding, Urease Inhibition, and Anti-Brain-Tumor Activities"

_molecules, 2023, doi:10.3390/molecules28062707_

Round 1

Reviewer 1 Report

This manuscript deals with "Investigations on Newly Synthesized Bis-Acyl-Thiourea Derivatives of 4-Nitrobenzene-1,2-Diamine for Their DNA Binding, Urease Inhibition and Anti-Brain Tumor Activities". This article claims that using of Bis-Acyl-Thiourea Derivatives of 4-Nitrobenzene-1,2-Diamine could be a suitable Antitumor application . I suggest a minor correction and require a detailed clarification. Correction to be addressed by the authors as follows: The abstract is not well organized, where the sentences are incomplete and no continuity is there. It would be feasible, if include the significance of the current study in the abstract.

A brief description of how the authors selected information from the literature in the databases, as well as what time period they searched for, is missing.
Authors should justify and expand the information on the advantages and disadvantages of this synthetic material.
Authors should specify the main experimental conditions used on the evidences from the literature. Where they briefly describe the most important data reported in the literature in a homogeneous manner and sequence reinforcing the relevance of this agent as novel alternative.
Authors should discuss whether the use of this agent represents a solid alternative to existing commercial drugs. Also please discuss about the anticancer role on mitochondria.

Please compare possible anticancer effects with other potent anticancer drugs.
Please add below studies to your manuscript in discussion section using below manuscripts:
DOI: 10.1155/2021/4946711

DOI: 10.1016/j.chemosphere.2022.134826

Conclusions should reaffirm the fundamental contribution of this paper.

Reviewer 2 Report

The manuscript has been reviewed and needs minor revision before publication:

1. At some places, English needs refinement.

2. Check line 468,UP-3> UP-1> UP-1 in conclusion.

3. Add HRMS for the synthesized compounds.

4. Add the electrochemical CV studies in abstract and conclusion.

Reviewer 3 Report

The authors have undertaken experimental investigations on newly synthesized bis-Acyl-Thiourea Deriv- 2  derivatives of 4-Nitrobenzene-1,2-Diamine for their DNA binding,  Urease Inhibition and Anti-Brain Tumor Activities, with support of combined quantum computational study using DFT electronic structure tools, and Molecular Docking calculations.  The study still interesting, but needs minor revision, more explanations with requested formatting changes, to be accepted.

1- Why did you use ADF for geometry optimization, when you used semi-empirical methods ' MOPAC.

2-  Why do you evaluate the energy of the interaction at each step of the simulation,

3- The conclusion is fairly low linked to the results and needs more development. 4- The authors have used ADF (Amsterdam Density Functional) software, which is currently a part of the AMS (Amsterdam Molecular Simulations) program, for quantum calculations and visualization, so you mentioned Gaussian and Gaussview in refs 40 and 41. Please chek it out. 5- The GGA: PBE/DZ level of theory is fairly poor for such systems, and DZP even TZP extended basis set are highly recommended. 6- Figure 5 should be resized, unclear items.

Round 2

Reviewer 4 Report

Report molecules-2248173 

Authors have addressed some of the observations of the first report. Nevertheless, some critical issues are still without solution. The following issues must be solved before publication of the computational results of the present manuscript. If authors did not solve these issues, I do recommend eliminating the computational section of this manuscript.  

Major points: 

  1. The confirmation of a minimum, using a frequency calculation, of an optimized geometry is a standard procedure in computational chemistry. The results of frontier orbitals and conceptual DFT descriptors of molecules without such confirmation are not reliable. Authors answers suggest a misunderstanding regarding the physical meaning of a frequency calculation, it is not used for the search of excited states. Moreover, it is needed for the confirmation of an equilibrium geometry of the ground state.  

  1. This point is related to the first one, an exploration of the Potential Energy Surface ensures that the computed chemical descriptors are those for molecules with the lowest energies, that is its global minimum. I do encourage authors to explore all the possible minimums, not the transition states, to get computational results related with the lowest energy conformations. Again, the author's answers suggest a misunderstanding regarding the physical meaning of a frequency calculation. 

  1. I encourage authors to perform a Docking “control” with Doxorubicin, there are co-crystalized structures of this drug with DNA (for instance: DOI: 10.2210/pdb215D/pdb), this could be a nice validation of the docking procedure for this work, with the possibility pf getting less than 2 Angstroms of RMSD.
